# First Tarsometatarsal Joint Fusion in Foot—A Biomechanical Human Anatomical Specimen Analysis with Use of Low-Profile Nitinol Staples Acting as Continuous Compression Implants

**DOI:** 10.3390/medicina59071310

**Published:** 2023-07-15

**Authors:** Andrew Sands, Ivan Zderic, Michael Swords, Dominic Gehweiler, Daniel Ciric, Christoph Roth, Christoph Nötzli, Boyko Gueorguiev

**Affiliations:** 1New York–Presbyterian Lower Manhattan Hospital, New York, NY 10038, USA; aksands@gmail.com; 2AO Research Institute Davos, 7270 Davos, Switzerland; dominic.gehweiler@aofoundation.org (D.G.); daniel.ciric@aofoundation.org (D.C.); boyko.gueorguiev@aofoundation.org (B.G.); 3Michigan Orthopedic Center, Lansing, MI 48911, USA; foot.trauma@gmail.com; 4DePuy Synthes, West Chester, PA 19380, USA; croth9@its.jnj.com; 5AO Foundation, 7270 Davos, Switzerland; christoph.noetzli@aofoundation.org

**Keywords:** arthrodesis, biomechanics, continuous compression implant, crossed screws, deformity, first tarsometatarsal joint, fixation, fusion, human anatomical cadaveric model, locking plate, nitinol staple, stabilization

## Abstract

*Background and Objectives*: The aim of this study was to investigate under dynamic loading the potential biomechanical benefit of simulated first tarsometatarsal (TMT-1) fusion with low-profile superelastic nitinol staples used as continuous compression implants (CCIs) in two different configurations in comparison to crossed screws and locked plating in a human anatomical model. *Materials and Methods*: Thirty-two paired human anatomical lower legs were randomized to four groups for TMT-1 treatment via: (1) crossed-screws fixation with two 4.0 mm fully threaded lag screws; (2) plate-and-screw fixation with a 4.0 mm standard fully threaded cortex screw, inserted axially in lag fashion, and a 6-hole TMT-1 Variable-Angle (VA) Fusion Plate 2.4/2.7; (3) CCI fixation with two two-leg staples placed orthogonally to each other; (4) CCI fixation with one two-leg staple and one four-leg staple placed orthogonally to each other. Each specimen was biomechanically tested simulating forefoot weightbearing on the toes and metatarsals. The testing was performed at 35–37 °C under progressively increasing cyclic axial loading until construct failure, accompanied by motion tracking capturing movements in the joints. *Results*: Combined adduction and dorsiflexion movement of the TMT-1 joint in unloaded foot condition was associated with no significant differences among all pairs of groups (*p* ≥ 0.128). In contrast, the amplitude of this movement between unloaded and loaded foot conditions within each cycle was significantly bigger for the two CCI fixation techniques compared to both crossed-screws and plate-and-screw techniques (*p* ≤ 0.041). No significant differences were detected between the two CCI fixation techniques, as well as between the crossed-screws and plate-and-screw techniques (*p* ≥ 0.493) for this parameter of interest. Furthermore, displacements at the dorsal and plantar aspects of the TMT-1 joint in unloaded foot condition, together with their amplitudes, did not differ significantly among all pairs of groups (*p* ≥ 0.224). *Conclusions*: The low-profile superelastic nitinol staples demonstrate comparable biomechanical performance to established crossed-screws and plate-and-screw techniques applied for fusion of the first tarsometatarsal joint.

## 1. Introduction

The first tarsometatarsal (TMT-1) joint—incorporating a trapezoid-shaped plane—is described as a ‘helicoid’ subtly complex anatomic configuration. Under physiologic conditions, it is well constrained by soft-tissue structures allowing a certain amount of up-and-down and rotational movements [1,2]. However, in an advanced foot deformity, a considerable increase in the first intermetatarsal angle is present, along with hypermobility and instability of the joint [1,3,4]. Corrective osteotomy with arthrodesis is an established technique for management of deformity and arthrosis of different origins [1,5,6,7,8,9]. Historically, different modifications from the original technique for Lapidus arthrodesis have evolved [10].

Currently, the standard TMT-1 fusion technique is fixation with two or three screws inserted in a lag fashion. One of the screws is inserted from the central dorsal aspect of the first metatarsal and routed to the plantar medial aspect of the medial cuneiform. The second screw is inserted from the dorsal aspect of the medial cuneiform and routed laterally to the first screw in the first metatarsal, exiting the plantar cortex. The two screws cross at a point on the longitudinal axis of the first metatarsal. This fixation has the disadvantage of involving 6–8 weeks immobilization [1,11]. Moreover, implant failure rates of up to 4.5% and non-union in up to 15% of the cases are reported [7,8,12]. Use of locked plates as an alternative fixation technique could conceivably allow for earlier weightbearing and improve bony union [1]. However, previous investigations have reported controversial findings [6,13,14,15].

On the other hand, nitinol devices (Figure 1) are part of a broader category of metallic staples used clinically for fracture fixation, arthrodesis and corrective osteotomies [16]. They are made of an alloy of nickel and titanium—a material exhibiting unique shape memory and superelastic characteristics based on a reversible solid–solid phase transformation from a highly ordered austenitic crystal to a less ordered martensitic structure [17,18]. Due to their superelastic characteristics, modern nitinol devices allow for an intraoperative interfragmentary and intersegmental compression with the ability of elastic recovery from large deformations which can occur in vivo, thus imparting a continuous dynamic compressive capability in contrast to conventional (non-nitinol) staples, plates and screws. This has been demonstrated in previous biomechanical studies reporting more compression achieved with nitinol devices versus conventional staples, along with a higher resistance to permanent deformation and a full shape recovery after loading [19,20,21,22,23].

Clinically, the ability of the nitinol devices to act as continuous compression implants (CCIs) by applying and maintaining a uniform active compression offers an attractive alternative to conventional screw and plate TMT-1 fusion. Moreover, in contrast to screw fixation that occupies a valuable area across the joint interface, the CCIs could allow full joint coaptation by maximizing the footprint for fusion. Although experience was gained with the use of CCIs in foot surgery [24,25,26,27,28], the clinical research evaluating their applications for TMT-1 fusion is still scant. So far, there is only one existing investigation reporting a non-union rate of 8.3% after treatment with two types of nitinol devices in 36 cases [29], which is well within the range of other methods for arthrodesis [7,8,30,31,32].

Therefore, the aim of this study was to investigate under dynamic loading the potential biomechanical benefit of simulated TMT-1 fusion with CCIs in two different configurations in comparison to crossed screws and locked plating in a human anatomical model. It was hypothesized that the CCIs would allow more elastic intersegmental movements compared with the other arthrodesis techniques.

## 2. Materials and Methods

### 2.1. Specimens and Preparation

Sixteen pairs of fresh-frozen (−20 °C) human anatomical lower legs from nine male and seven female donors (mean age 74.5 years, range 61–95 years) with no history of pathology were used in this study carried out in 2019. All donors gave their informed consent within the donation of an anatomical gift statement during their lifetime. The specimens were thawed at room temperature for 24 h prior to preparation and biomechanical testing.

Bone mineral density (BMD) was measured in the calcaneus via computed tomography (CT) scanning (Revolution EVO, GE Medical Systems AG, Opfikon, Switzerland) at a slice thickness of 0.63 mm with the use of a calibration phantom (European Forearm Phantom QRM-BDC/6, QRM GmbH, Möhrendorf, Germany). The region of interest was a cylinder of 15 mm diameter and 25 mm length, centrally located and oriented longitudinally from the calcaneo-cuboidal joint to the calcaneal tuberosity, at a distance of 10 mm from the latter.

Based on BMD, the specimens were randomized in two clusters of eight pairs each. The lower legs in each cluster were further assigned pairwise for either TMT-1 crossed-screws fixation (Group 1) or plate-and-screw fixation (Group 2), CCI fixation with two two-leg nitinol devices (Group 3) or CCI fixation with one two-leg nitinol device and one four-leg nitinol device (Group 4, Figure 2). Each group consisted of equal right and left specimens. The sample size of *n* = 8 was chosen based on a priori power analysis with the assumption that the standard deviation in each group is not larger than 70% of the minimum difference in mean values between the groups. The lower legs were well randomized to the four groups, with BMD being not significantly different between them (*p* = 0.922), as shown in Table 1. Therefore, a fair start condition was assured for each group.

All procedures were performed on intact feet with preparation of the joint articulations and under fluoroscopic control according to the implant manufacturer’s guidelines.

A dorsal approach to the midfoot and the metatarsals was performed, taking care not to damage the tibialis anterior and posterior tendons. An experienced surgeon carried out all instrumentations, thus ensuring consistency of the procedures. Provisional reduction was performed with pointed forceps and temporary Kirschner (K-) wires inserted across the joint. All implants were provided by the same manufacturer (DePuy Synthes, West Chester, PA, USA).

Crossed-screws fixation was performed with two 4.0 mm fully threaded stainless steel lag screws inserted in standard fashion, i.e., the first screw from first metatarsal to plantar medial cuneiform and the second screw from medial cuneiform to the plantar base of the first metatarsal. Pilot holes were drilled in a 4.0/2.5 mm combination to achieve lag screw fixation.

Plate-and-screw fixation commenced with insertion of a 4.0 mm standard midfoot fully threaded stainless steel cortex screw, inserted cortically in lag fashion, followed by fixation with a 6-hole TMT-1 stainless steel Variable Angle (VA) LCP Fusion Plate 2.4/2.7 of 39 mm standard length. The plate was positioned dorsomedially on the medial cuneiform and the first metatarsal via olive wires. Pilot holes were drilled using dedicated 2.0 mm VA Locking Drill Guides and drill bits of the same dimensions. Drill hole trajectories were chosen individually for optimal screw purchase in the bone. All 6 plate holes were occupied with 2.7 mm VA stainless steel locking screws of sufficient screw length, allowing for additional transfixation of the intermediate cuneiform and second metatarsal bones. All screws were locked at 1.2 Nm torque.

CCI simulated fusion with two two-leg nitinol devices was performed with an SE-1818TI DPS BME SPEEDTITAN™ and an SE-2520TI DPS BME SPEEDTITAN™ implant placed at 45° angulation dorsolaterally and dorsomedially, respectively. Pilot holes were drilled in a bicortical fashion using a dedicated drill guide and a 2.65 mm drill bit. The nitinol devices were inserted and released using an insertion device to standardize the insertion process and prevent any deformation of the nitinol devices at the time of insertion.

CCI simulated fusion with a two-leg nitinol device and a four-leg nitinol device was performed with an EL-1818S2 DPS BME ELITE^®^ and an EL-2520S4 DPS BME ELITE^®^ implant placed at 45° angulation dorsolaterally and dorsomedially, respectively. Pilot holes were predrilled in a bicortical fashion using a drill guide and a 3.0 mm drill bit.

All specimens were denuded of all soft tissues in the proximal 80 mm of the tibia and fibula and their proximal end was embedded in a 65 mm long cylindrical form of 48 mm diameter using polymethylmethacrylate (SCS-Beracryl, Suter Kunststoffe AG, Jegenstorf, Switzerland). Distally to the embedding, a 2 cm wide stripe of soft tissue around the Achilles tendon was removed up to the level of the calcaneus to allow insertion of a metallic eyebolt through the calcaneus, used as an anchor for simulated Achilles tendon fixation. Finally, retro-reflective marker sets were attached to the distal tibia, talus, navicular, medial cuneiform and first metatarsal for motion tracking.

### 2.2. Biomechanical Testing

Biomechanical testing was performed on a servo-hydraulic material testing system (MTS Bionix 858; MTS Systems Corporation, Eden Prairie, MN, USA) equipped with a 4 kN load cell.

A room temperature of 35–37 °C—the same as in the human body—was ensured with a temperature-sensing fan-heater. The test setup simulated forefoot weightbearing (Figure 3). The proximal embedding was fixed to the machine actuator. The toes were placed under fluoroscopic control on a custom-made radiolucent wooden platform. The calcaneus was connected to the machine actuator via a metallic turnbuckle, anchored to an eyebolt inserted in the calcaneus and another eyebolt mounted on the transducer flange. The initial foot position was adjusted with the turnbuckle so that the angle between the first metatarsal and the horizontal line in mediolateral view was 20° under 20 N axial preload.

Biomechanical testing commenced with initial quasi-static axial ramped loading from 20 N to 150 N at a rate of 13 N/s, followed by progressively increasing cyclic axial loading at 2 Hz with a physiologic profile of each cycle [33]. Whereas the valley cyclic load was held at a constant level of 20 N, the peak load, starting from 150 N, increased at a rate of 0.08 N/cycle until 100 mm axial displacement of the machine actuator, which was considered large enough to provoke catastrophic failure of the tested constructs.

### 2.3. Data Acquisition and Evaluation

Machine data in terms of axial load and displacement were acquired at 128 Hz. Based on these, initial construct stiffness was calculated from the ascending slope of the load–displacement curve during initial quasi-static ramped loading. Mediolateral X-rays were acquired every 250 cycles under peak loading using a triggered C-arm (Siemens Arcadis Varic, Siemens AG, Erlangen, Germany) for visual examination of any temporal degradation process within the bone–implant interface.

Interfragmentary movements were captured at 100 Hz by motion tracking in all six degrees of freedom using five infrared cameras (MCU120, Qualysis AB, Gothenburg, Sweden) operating at an accuracy of 1.4–2.4% and a precision of 1.9–3.9 µm [34]. Based on these data, combined angular deformation between the two bones of the TMT-1 joint in both adduction and dorsiflexion (gap angle) was calculated together with the displacements at the dorsal and plantar aspects of the joint after 500, 1000 and 2500 cycles under peak (loaded condition) and valley (unloaded condition) loading.

Statistical analysis was performed using SPSS software package (v.27, IBM SPSS, Armonk, NY, USA). The normality of data distribution was screened and proved using the Shapiro–Wilk test. Paired-Samples *t*-test and General Linear Model Repeated Measures with Bonferroni Post Hoc tests for multiple comparisons were conducted to identify significant differences between the fixation techniques. The level of significance was set to 0.05 for all statistical tests.

## 3. Results

The descriptive data of all outcome measures are summarized in Table 1.

Initial stiffness of the specimens did not differ significantly among the four fixation techniques (*p* = 0.872).

Gap angle—i.e., the combined angular deformation between the two bones of the TMT-1 joint in both adduction and dorsiflexion—in unloaded foot condition increased significantly between 500 and 2500 cycles in each group (*p* < 0.001); however, no significant differences were observed among all pairs of groups (*p* ≥ 0.128). In contrast, the amplitude of this deformation between unloaded and loaded foot conditions within each cycle also increased significantly between 500 and 2500 cycles in each group (*p* < 0.001) but was additionally significantly bigger for the two CCI fixation techniques compared to both crossed-screws and plate-and-screw techniques (*p* ≤ 0.041). No significant differences were detected between the two CCI techniques, as well as between the crossed-screws and plate-and-screw techniques (*p* ≥ 0.493) for this parameter of interest.

Displacements at the dorsal and plantar aspects of the TMT-1 joint in the unloaded foot condition, together with their amplitudes between unloaded and loaded foot conditions within each cycle, increased significantly between 500 and 2500 cycles in each group (*p* ≤ 0.04) but did not differ significantly among all pairs of groups (*p* ≥ 0.224).

Specimens in all treatment groups failed at the fixation site in approximately 50% of the cases, whereas in the other 50% the site of failure was expressed by either a fixation failure of the calcaneus or general plantar gap opening of the whole foot. At the fixation site, specimens failed by a distinct gap opening and concomitant fragment displacement. This was ascribed either to implant loosening or its cut-through through the bone. Within Group 4, there were two specimens that had a fracture of the medial cuneiform, and one specimen with a fracture of the first metatarsal (Figure 4). No implant failures were detected in any of the specimens.

## 4. Discussion

The aim of this study was to investigate under dynamic cyclic loading the potential biomechanical benefit of a fixation with nitinol devices in different configurations in comparison to cross-screws and plate-and-screw fixations in a TMT-1 fusion model.

The tested hypothesis could only be partially confirmed to the extent that TMT-1 fixation with nitinol devices results in more flexible constructs compared to a conventional technique using either crossed screws or a plate, which was expressed by a larger amplitude of the intersegmental movements between the loaded and unloaded states of the feet. This is a logical consequence of the type of fixation methods. On the one hand, the trajectory of the crossed screws followed a path lying within the bony structures, ensuring a low lever arm relative to the load application point. Furthermore, it allowed joint compression by applying the lag-screw technique, enhancing the resistance against shear movements. As a result, crossed-screws fixation can shield external loads and efficiently protect the construct against bending and shearing. On the other hand, plate fixation uses a relatively strong plate which is attached to the dorsomedial aspect of the TMT-1 joint. This renders the construct resistive against the plantar gap opening due to the high moment of inertia provided by the plate aligned in the bending plane. Consequently, the relatively low-profile nitinol devices inserted on the dorsal aspect of the joint turned out to be the weakest constructs.

The part of the tested hypothesis that the nitinol devices would demonstrate lower residual plastic deformations in TMT-1 fusion under unloading conditions over time needed to be rejected. The observation was made that the residual displacements were uniformly distributed across all fixation techniques in terms of statistical significance. Hence, the theory of superelasticity may not be transferred one-to-one in a biomechanical setup using human anatomic preparations. The interpretation of this finding may be multifactorial. Main reasons could be the weak holding strength of the staple legs in the bone, leading to early onset of micro-movements and their accompanying loosening, and the potentially weak spring-effect of the superelasticity feature.

The present findings stand somewhat in contradiction to previous investigations using the same type of staples. In their study, Russell et al. [23] investigated the biomechanical performance of two different staple configurations in a simulated Lapidus procedure using artificial foot models and found that, whereas fixation with two staples arranged orthogonally to each other provides overall higher biomechanical outcome measures, including contact pressure within the joint footprint under different loading scenarios compared to a single staple fixation, both fixation groups fully recovered the plantar gapping and restored the initial joint configuration. In a subsequent study performed by Aiyer et al. [21], the same foot models were tested under four-point bending for biomechanical comparison of Lapidus arthrodesis constructs fixed with either one conventional staple, two conventional staples, two crossed screws, or a claw plate. Similar outcomes were detected for the two conventional staple techniques, which demonstrated sustained compression within the fixed joint in contrast to both fixations with screws and a claw plate. The authors argued that the multiplanar fixation achieved with two staples may even facilitate earlier postoperative weightbearing due to improved stability. A further coherent study by Hoon et al. [22] went on to compare one- and two-staple fixation versus plate fixation in standardized foam blocks under multiplanar loading. Again, whereas both staple techniques tolerated less loads until certain gap displacement, they exerted a significantly higher compressive force across the osteotomized contact area and formed a significantly smaller permanent gap compared to plate fixation.

Disregarding their main arguments, the authors of all three above-mentioned studies also discuss the alternative use of staples as adjunctive fixation devices to standard fixation methods rather than stand-alone implants to impart compression across the joint. However, the benefit of such a combined use still needs to be demonstrated in vitro.

Nitinol devices attracted attention in biomechanical studies striving for alternative fixation techniques in other anatomical body sites, such as for clavicle or patella fractures [35,36,37]. Whereas a good resistance against articular gap opening was observed in the patella, the results of clavicle fracture fixation were rather mediocre. In summary, biomechanical studies on nitinol devices are inconclusive.

On the other hand, clinical studies using nitinol devices for Lapidus procedures showed promising results in terms of union rates [24,29]. However, in those studies patients were either allowed to bear weight in rigid shoes or not at all. This suggests that staples acted rather as static than dynamic stabilizers. Historically, nitinol devices acting as CCIs have already been successfully used not only in foot surgery [28,38] but also in other specialties, such as the spine [39,40].

In the current study, nitinol devices were placed at the dorsal compression site of the foot to allow continuous compression across the joint while unloading the foot. From this perspective and in alignment with other studies investigating plating of the medial column [33], although a plantar nitinol device placement would be located on the tension site and provide enhanced resistance against plantar gapping, it would conflict with the dogma of superelasticity provided by the nitinol devices and risk injury and irritation to the anterior tibialis.

A main limitation of the current study was the use of anatomical specimens and the associated disabled active stabilizing contribution of muscles and ligaments to the simulated walking cycle. As a compensation for it, the Achilles tendon was passively activated via its attachment to the machine transducer. Although it remains uncertain to what extent this force surrogate is beneficial for stressing the fixation constructs, the observed failure modes render the used setup encouraging. Another limitation of the study was the use of intact specimens instead of specifically ones with deformity indicated for TMT-1 fusion. Collection of such specimens would go beyond the scope of practicability and also induce unknown factors due to variability in foot deformation. Finally, only the propulsion section of a gait cycle could be simulated neglecting the more complex stress on the foot while simulating the full gait cycle. Hence, different results may be expected under different loading modes. Related to this, the study was underpowered within the meaning of high data scattering due to diluted focus on the fixation site for the sake of more physiological gait simulation.

The advantage of the present study was the use of precise motion tracking, enabling continuous and accurate measurement of intersegmental deformation during the whole cyclic tests. Furthermore, the selected parameters for stability assessment deemed clinically relevant and were also considered in previous studies [6,14,41]. Finally, the test protocol was chosen aggressively enough to provoke failure in the constructs and allowed stressing the constructs in their worst-case scenario.

Future studies shall focus on whether a stand-alone use of staples represents a sufficient treatment or whether the concept must be reconsidered towards their use as adjunctive implants providing continuous compression across the joint, whereas plates and screws take over the main part as external load carriers.

## 5. Conclusions

From a biomechanical perspective, based on human anatomical setting, the low-profile superelastic nitinol staples, used as continuous compression implants for more elastic stabilization under active compression, demonstrate comparable performance to established crossed-screws and plate-and-screw techniques applied for fusion of the first tarsometatarsal joint and allow full joint coaptation by maximizing the footprint for fusion.

## Figures and Tables

**Figure 1 medicina-59-01310-f001:**
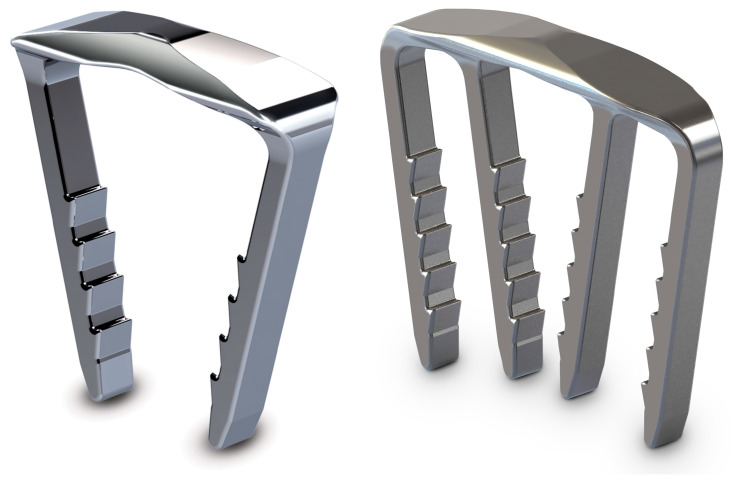
Two-leg (**left**) and four-leg (**right**) nitinol devices from the series DPS BME ELITE^®^ (DePuy Synthes, West Chester, PA, USA).

**Figure 2 medicina-59-01310-f002:**
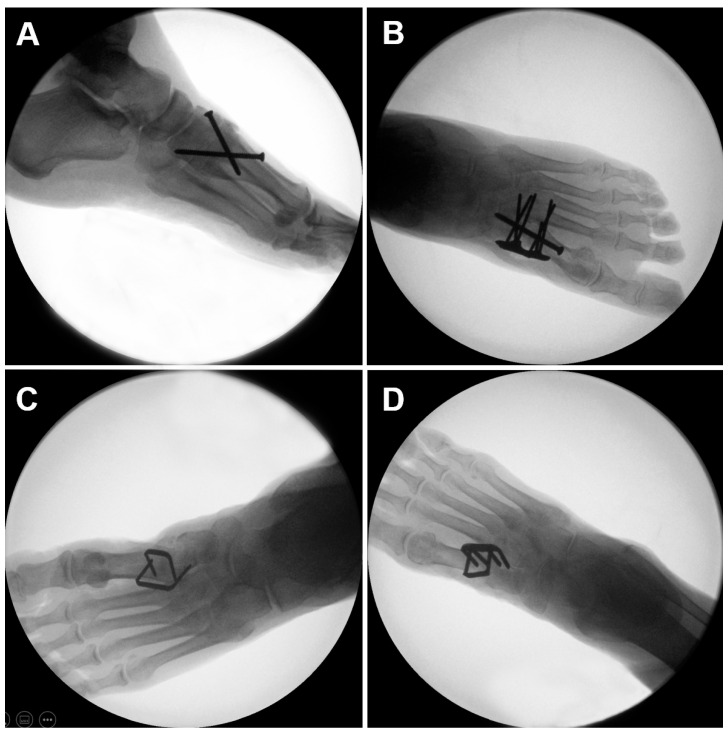
Radiographs visualizing fixation of the TMT-1 joint with use of crossed screws (**A**), plate-and-screw technique (**B**), two nitinol devices from the series DPS BME SPEEDTITAN™ (DePuy Synthes, West Chester, PA, USA) (**C**), and two nitinol devices from the series DPS BME ELITE^®^ (DePuy Synthes, West Chester, PA, USA) (**D**).

**Figure 3 medicina-59-01310-f003:**
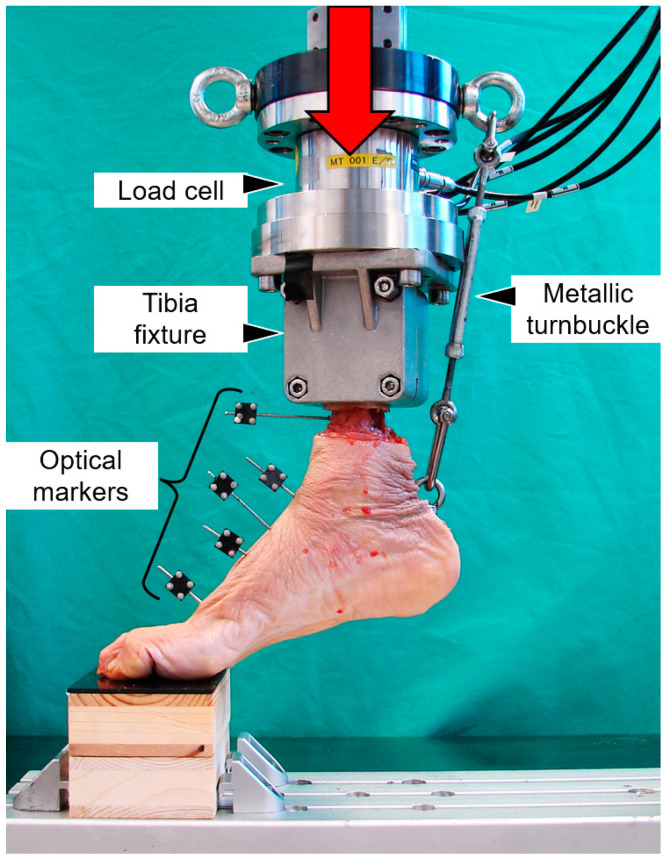
Setup with a specimen mounted for biomechanical testing and equipped with optical markers for motion tracking, attached to the distal tibia, talus, navicular, medial cuneiform and first metatarsal.

**Figure 4 medicina-59-01310-f004:**
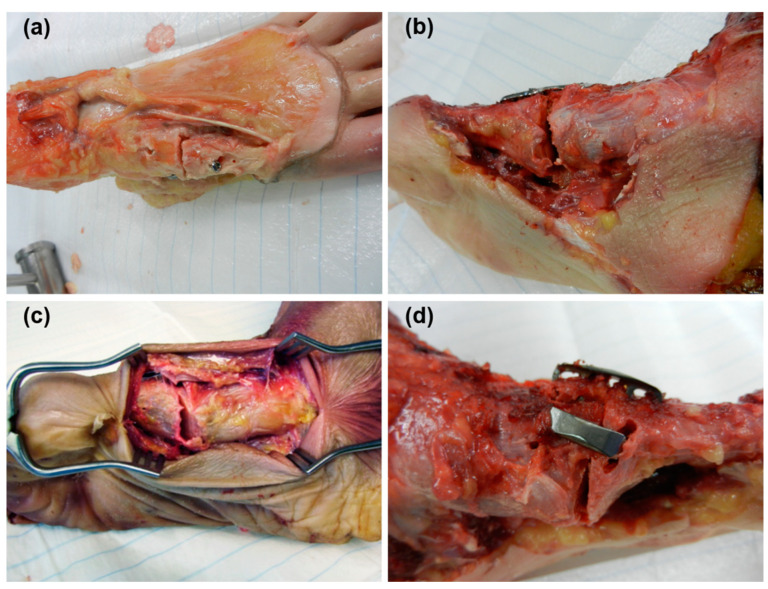
Photographs of specimens depicting exemplified failure modes following (**a**) crossed-screws fixation (displacement and screw cut-through); (**b**) plate-and-screw fixation (gap opening); (**c**) CCI simulated fusion with two two-leg nitinol devices (gap opening and fragment displacement); (**d**) CCI fusion with one two-leg and one four-leg nitinol device (fractured cuneiform, cut-through of the medial nitinol device and gap opening).

**Table 1 medicina-59-01310-t001:** Parameters of interest in each separate group presented in terms of mean value and standard deviation.

Parameter		Groups			
		1	2	3	4
BMD (mgHA/cm^3^)		123.5 (48.5)	111.8 (41.3)	119.7 (48.23)	108.4 (49.9)
Initial stiffness (N/mm)		18.1 (4.0)	19.0 (5.3)	19.6 (7.7)	20.2 (3.7)
	Cycles				
Gap angle unloaded (°)	500	0.05 (0.10)	0.00 (0.01)	0.13 (0.17)	0.19 (0.42)
	1000	0.06 (0.13)	0.04 (0.07)	0.41 (0.48)	0.40 (0.61)
	2500	0.16 (0.25)	0.14 (0.24)	0.99 (0.90)	0.70 (0.84)
Gap angle amplitude (°)	500	0.56 (0.12)	0.53 (0.09)	1.00 (0.22)	0.88 (0.46)
	1000	0.58 (0.14)	0.61 (0.20)	1.14 (0.21)	0.92 (0.55)
	2500	0.68 (0.19)	0.66 (0.28)	1.80 (0.98)	1.31 (0.82)
Displacement at dorsal aspect unloaded (mm)	500	0.02 (0.07)	0.01 (0.05)	0.04 (0.05)	0.11 (0.16)
	1000	0.04 (0.10)	0.06 (0.10)	0.08 (0.10)	0.19 (0.29)
	2500	0.09 (0.11)	0.14 (0.23)	0.19 (0.27)	0.32 (0.34)
Displacement amplitude at dorsal aspect (mm)	500	0.32 (0.10)	0.33 (0.08)	0.41 (0.22)	0.39 (0.20)
	1000	0.32 (0.11)	0.36 (0.09)	0.46 (0.29)	0.45 (0.32)
	2500	0.38 (0.12)	0.37 (0.07)	0.71 (0.69)	0.48 (0.27)
Displacement at plantar aspect unloaded (mm)	500	0.02 (0.07)	0.01 (0.05)	0.07 (0.11)	0.07 (0.10)
	1000	0.05 (0.10)	0.10 (0.11)	0.16 (0.19)	0.14 (0.19)
	2500	0.12 (0.10)	0.13 (0.21)	0.50 (0.67)	0.29 (0.26)
Displacement amplitude at plantar aspect (mm)	500	0.47 (0.15)	0.49 (0.10)	0.77 (0.34)	0.61 (0.31)
	1000	0.50 (0.19)	0.86 (1.00)	0.84 (0.51)	0.75 (0.52)
	2500	0.57 (0.22)	0.67 (0.28)	1.29 (1.14)	0.76 (0.49)

## Data Availability

The datasets used and/or analyzed during the current study are available from the corresponding author on reasonable request.

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
