# Peer review of "First Tarsometatarsal Joint Fusion in Foot—A Biomechanical Human Anatomical Specimen Analysis with Use of Low-Profile Nitinol Staples Acting as Continuous Compression Implants"

_medicina, 2023, doi:10.3390/medicina59071310_

Round 1
Reviewer 1 Report
The manuscript is well-written with sufficient illustration.
I suggest the authors to mention the "staples" in the title, since the terms "CCI" may be not familiar enough to the general readers
Author Response
Dear Reviewer
Thank you for your time dedicated to critically review our manuscript.
Please find attached our reply, addressing each request point by point.

Reviewer 2 Report
I think it is an excellent research work, mainly aimed at looking at the properties of a material. However, I believe the context should be more inclined towards clinical research, orthopedic in this case. We are really testing the characteristics of a material that has already been tested in other studies. We do not know its clinical implications. It is true that it has special biomechanical properties, but we are not assessing clinical results at any time.
Introduction.- The problem is well-focused, and the objective of the study is well-stated, although perhaps too much focused on the properties of the material rather than on clinical results.
Materials and Methods: I understand that it is very difficult to have these anatomical models, but there is no previous design, no minimum number of cases, no evaluation of the variable to be studied. In Ln 110 it is said that it is randomized according to BMD, although in the results it is said that there are no differences in this variable. I would like to know what is the assignment criteria for the groups.
Discussion: I think that the limitations are very appropriate, especially considering that we are dealing with intact models, which do not reproduce the muscular and ligamentous contribution to joint stability.
Author Response

(The authors gave the same response as above.)

Reviewer 3 Report
The manuscript is of excellent quality, meticulously written and detailed in every section.
Images and tables included make understanding of both methodological approach and results fluent and intuitive.
Conclusions are in agreement with the results.
The manuscript is worthy of publication.
Author Response

(The authors gave the same response as above.)

Round 2
Reviewer 2 Report
I believe the corrections are adequate and sufficient.